# Life beyond Loss: A Retrospective Analysis of the Impact of Meaning of Life Therapy on the Grieving Process of Cancer Patients’ Family Caregivers

**DOI:** 10.3390/healthcare12040471

**Published:** 2024-02-14

**Authors:** Maria João Freitas, Sónia Remondes-Costa, Elisa Veiga, Gerly Macedo, Ricardo João Teixeira, Manuela Leite

**Affiliations:** 1Department of Social and Behavioural Sciences, University Institute of Health Sciences (IUCS-CESPU), 4585-116 Gandra, Portugal; a26045@alunos.cespu.pt; 2Department of Education and Psychology, University of Trás-os-Montes e Alto Douro, 5000-622 Vila Real, Portugal; costas@utad.pt; 3Research Centre for Human Development, Faculty of Education and Psychology, Universidade Católica Portuguesa, 4169-005 Porto, Portugal; eveiga@ucp.pt; 4Clinical and Health Psychology Unit, Psychiatry and Mental Health Service, Hospital da Senhora da Oliveira, 4835-044 Guimarães, Portugal; gerlymacedo@hospitaldeguimaraes.min-saude.pt; 5REACH—Mental Health Clinic, 4000-138 Porto, Portugal; ricardo@reach.com.pt; 6CINEICC (Center for Research in Neuropsychology and Cognitive and Behavioral Intervention), Faculty of Psychology and Education Sciences, University of Coimbra, 3004-531 Coimbra, Portugal; 7iHealth4Well-Being—Innovation in Health and Well-Being—Research Unit, Instituto Politécnico de Saúde do Norte, CESPU, 4560-462 Penafiel, Portugal

**Keywords:** family caregivers, cancer, palliative care, meaning of life therapy, grieving

## Abstract

Oncological disease in the palliative stage is a huge challenge for patients and their family caregivers (FCs) due to the fact that it confronts them with death, as well as physical, psychological, and existential suffering. Meaning of Life Therapy (MLT) is a brief structured psycho-existential intervention aiming to help patients in a meaning-making life review process, promoting end-of-life adaptation. The Life Letter (LL) resulting from MLT is an element that facilitates communication between the patient and their caregivers. The goal of this study was to understand the impact of MLT on the grieving processes of eight FCs and to study their perceptions of the role of the LL on grief through semi-structured interviews. The results of our qualitative analysis indicate that MLT was perceived by the FCs as a positive experience despite the conspiracy of silence being identified as a drawback. The LL was interpreted as a communicational element, promoting emotional closeness with the cancer patients and serving as a valuable tool in the FCs’ adaptation to loss. Our research findings show that the needs of FCs, especially after experiencing the loss of their relative, are dynamic and specific. This is why it is urgent to develop interventions that consider the idiosyncrasies of end-of-life cancer patients and their FCs in order to avoid frustrated farewells, lonely deaths, and maladaptive grieving processes. This is the direction in which MLT should evolve.

## 1. Introduction

The diagnosis of an advanced, incurable, and progressive cancer disease significantly affects both patients and their families [1,2], frequently prompting emotional reactions such as sadness, anxiety [3], depression [4], guilt, anger, impotence, uncertainty, and fear [5]. When faced with this reality, both the patient and their family are forced to (re)connect and (re)normalize as ways to bring a new meaning to this life event, living on a flux that, usually, leads them to confront human uncertainty [6], implicating an incessant ability to adapt [5,7].

Although death is inherent to the human condition, patients and family members demonstrate communication difficulties [8], often resorting to the conspiracy of silence, a strategy to avoid discussing the patient’s health condition, prognosis, and finitude [9], fearing the repercussions that a knowledge of that information may have [10]. However, the existence of clear, open, and coherent communication [11] that serves to inform the patient about their diagnosis and prognosis [7] is essential, allowing the patient and family members to plan end-of-life care in advance together with the medical team [12]. Such care is based on an interactional communication flow between the caregiver triad: patient, family members, and health professionals [13].

The impact of a terminal cancer disease and the successive losses lead the patient to a state of profound vulnerability and fragility [14], with psychosocial, emotional, physical, and existential consequences [15,16]. The increased need for psycho-emotional and biopsychosocial-spiritual assistance [5,17,18] is reflected in the search for emotional relief [7,12] in a safe environment that favors open communication about end-of-life care [19], providing a proactive response [20] tailored to the patient’s needs. Coping with a terminal illness [21,22] is a subjective and complex experience [23] that involves a process of acceptance or positive reinterpretation. Otherwise, high levels of emotional suffering and frustration can occur due to the false expectation of improvement and recovery [11,21].

Family caregivers (FCs) also face emotional challenges concerning adaptation to this new reality, the physical deterioration of their loved one, and the confrontation with imminent death. Caregiving involves new responsibilities and skills [24,25] and is a physically and psychologically demanding role, especially at the end-of-life stage, which can result in caregiver burden and anticipatory grief [26,27]. It is also an opportunity for personal growth and a sense of accomplishment, having a positive impact on the reaction to loss and the grieving process [28].

This time between life and death, along with life consciousness vs. death awareness [29] and a sense of existential annihilation [30], is also the right time for interpersonal and personal growth [22,31], benefits that can be gleaned from psychological interventions [29].

The psychological approaches commonly used in palliative care are interventions centered on meaning or life review and cognitive behavioral therapies [32]. Recently, different psychotherapeutic interventions, such as the short-term structured Life Review [33,34,35], Dignity Therapy (DT) [36,37,38,39], and Meaning-Centered Psychotherapy [40], have been developed to address the specific desires and needs of terminally ill patients. The aim is to optimize therapeutic results [22], reflecting important aspects of their lives, strengthening their identity and their sense of meaning, purpose [41], and continuity [42], often contributing to a decrease in emotional and existential suffering [43].

Meaning of Life Therapy (MLT), developed by our team [44], proposes an innovative, brief, and individualized psychotherapeutic intervention through the process of a shared life review guided by a 14-question protocolized interview carried out over four sessions, the content of which is recorded, transcribed, edited, and compiled. The final objective of MLT is the elaboration of a written document—‘Life Letter’, allowing for the materialization of a physical legacy, which can be delivered to the patient’s relatives or significant others, if this is desired. The focus is on leading the patient to find a sense of meaning and purpose in life and suffering, as well as preserving their dignity and identity. Open reflective questions with important messages are also included, such as happy moments or significant events the image by which that the patient would like to be remembered, love affirmations for their family, farewells, choices, decisions, regrets, concerns, needs, desires, wishes, pleas for forgiveness, apologies, reconciliation, the resolution of outstanding issues, and the transmission of future advice [44]. The main idea of this document is that the final transcript will remain, so that the recalling of the patient’s biographical memories lasts over time. In addition, it is expected that the Life Letter facilitates communication with the family members and the feeling of a continuous relationship, even after the patient’s death, especially if its recipient(s) interpret it as a source of comfort.

The action of health professionals does not cease at the moment of the patient’s death; their work continues after the patient’s death, as they monitor the patient’s family members [45], taking into account that grief is one of the most painful experiences in people’s lives [46,47,48]. The answer to this phenomenon is heterogeneous [47], challenging the family unit to face the patient’s death and to adjust without the presence of their loved one [13]. Therefore, the uniqueness of the process and the different experiences in the face of loss have a distinct impact which is not always consistent with an adaptation to loss.

Nevertheless, it appears that large investments are being made in monitoring patients and their relatives in the process of palliative and terminal cancer disease, with, however, a gap in the post-mortem follow-up of family caregivers. It is in this context that the present study appears as an opportunity to ‘give voice’ to the FCs of deceased cancer patients who previously enrolled in MLT, contributing to the understanding of the loss process and to guiding the development of studies within the scope of the impact of MLT, which is in line with what was suggested by Cardoso et al. (2023). The research question we intended to answer was as follows: “What is the impact of the MLT on the grieving process of the family caregivers of the deceased loved one enrolled in this intervention?”.

## 2. Materials and Methods

### 2.1. Objectives

This study was integrated into a longitudinal study, developed in the Palliative Care Outpatient Service of a hospital located in the northern region of Portugal, involving dyads of patients diagnosed with advanced cancer with no expectation of being cured existing among them and their family caregivers [44]. Patients were enrolled to partake in MLT and elaborated the ‘Life Letter’ (LL), which was subsequently delivered to the FCs. After a cancer patient’s death and in order to respond to our research question, we defined the following objectives: (1) to understand the FC´s perception of the impact of MLT on the patient’s adaptation to the end-of-life stage; (2) to know the FC´s perception of the impact of MLT on their adaptation to the end-of-life of their loved one; (3) to evaluate the role of the LL in the grieving process; and (4) to explore the aforementioned adaptation’s role in the grieving process.

### 2.2. Method

A qualitative research design was used, since qualitative methods provide a source of rich and well-found descriptions and explanations of a phenomenon [49] through the experiences of those who directly experienced them, recognizing and contextualizing the participant’s point of view [50]. Accordingly, the present study used semi-structured interviews as its main data collection method.

### 2.3. Participants

We used a carefully selected, purposive sample [51]. Participants were recruited according to the following inclusion criteria: (1) being a FC of a cancer patient that was enrolled in MLT; (2) the LL has already been received by FC; (3) the cancer patient has already passed away; and (4) the absence of a psychopathology and/or cognitive impairment that compromises one’s ability to respond to the interview.

Nineteen dyads (cancer patients and FCs) that were integrated our previous studies were initially identified. Three dyads did not fulfill the 3rd inclusion criterion and were therefore excluded from the study. The 16 FCs were contacted by phone, through which they were informed about the objectives of the study and invited to participate. Only 9 FCs fulfilled all the inclusion criteria and accepted the opportunity to take part in the study. However, one of the participants was excluded during the interview stage after revealing that they did not remember the LL and presenting an incoherent and divergent speech, making data collection unfeasible (inclusion criteria 4). The final sample consisted of 8 participants (Figure 1).

The participants were aged between 20 and 56 years old (*M* = 38.38; *SD* = 11.86), and the majority were females. The time that had elapsed since the death of their relative varied between 5 months and 4 years and 1 month (Table 1). Regarding the degree of kinship with the deceased, almost all participants lost their parents, followed by those that lost their spouses and grandmother.

### 2.4. Procedure

For data collection, a semi-structured interview script included 23 questions was elaborated in order to assess the FCs’ perceptions of the role of MLT in the patient’s death process (e.g., “What do you remember about MLT with your relative?”; “What do you think that were the positive and negative aspects of that intervention?”) and the role of the LL in their grieving process (e.g., “Did you read the LL?”; “What were your feelings and thoughts?”). They were also asked about how they dealt with their loss (e.g., “What has been the biggest difficulty for you after the death of your family member?”).

Data collection was carried out in person at a hospital located in the northern region of Portugal by a specific researcher, a psychologist of the Palliative Care Outpatient Service, between 11 April and 20 June 2022. The interviews were scheduled according to the participants’ convenience and were audio-recorded, with the participants giving their written informed consent for them to be recorded. They had an average duration of 30 min. This investigation was approved by the health ethics committee of a hospital in the north of Portugal (Scientific Project No. 107/2021 CAF).

### 2.5. Data Analysis

All the audio interviews were transcribed ipsis verbis into documents in Microsoft Word, assigning a code number to each participant. Global readings and exhaustive re-readings of all interviews were carried out in order to allow the researchers to become familiar with the data. In this process, information relevant to the investigation, such as meanings and thoughts, were identified. Analyses of the content the interviews were guided by a thematic analysis process [52]. The five steps inherent to the thematic analysis process—compiling, disassembling, reassembling, interpreting, and concluding—were simultaneously respected [53]. Therefore, the coding process was developed in progressive stages, proceeding with the organization of all data—compilation, separation, and extraction of semantic units or content with latent meaning [52]. Gradually, a system of meaningful categories was built, raised in the participants´ narratives and framed by our research objectives [54]. In practice, there were advances and setbacks and a constant comparative analysis of themes and subcategories [55].

A joint analysis and the final verification of the coherence between the themes and data were carried out through assessing the explicit and implicit meanings [56] and proceeding to the process of description, analysis, and interpretation.

Three researchers from the team participated in the coding process and the development of the category system, and regular meetings were held to discussions [57] and triangulations of the decisions that were made [49], ensuring the consistency and reliability of our analysis [54]. The data analysis process was supported by the use of NVIVO 13 software.

## 3. Results

Our analysis of the qualitative data resulted in five main themes: (1) Remembrance of the Intervention; (2) FCs’ Perception about Patients’ Adaptation to the End-of-Life Stage; (3) FCs’ Adaptation to the Patients’ End of Life; (4) The Role of the LL in the End-of-Life Adaptation Process; and (5) Grieving Process.

The presentation of the results was carried out in line with the proposed objectives. This paper presents tables for some themes and describes the categories and subcategories that emerged from the data—highlighted in bold—including the corresponding number of participants (P) and the number of references (Ref.^es^). Excerpts from the interviews illustrating an interesting point pertaining to a category or subcategory (in italics) are also included, as well the code corresponding to the participant who provided that excerpt/quote (e.g., P2).

### 3.1. Remembrance of the Intervention

When we asked participants about the **remembrance of the intervention**, all of them remembered that their loved one participated in the MLT (8P/8Ref.^es^). They specifically highlighted the **content of the questions** (3P/3Ref.^es^); the **letter** (2P/2Ref.^es^); the **sessions** (2P/2Ref.es); **psychological support** (1P/1Ref.^e^); and **accompanying** the family member **to appointments** (1P/1Ref.^e^), retaining a **positive memory** of the intervention (1P/1Ref.^e^).

### 3.2. FCs’ Perceptions about Patients’ Adaptation to the End-of-Life Stage

Exploring the perceptions of the FCs about how MLT contributed to patients’ adaptation to the end-of-life stage (Table 2), most of the participants noted that MLT had a **positive** impact (6P/6Ref.^es^) since it facilitates **prognosis acknowledgment** (3P/4Ref.^es^). They referred to the intervention as a **private space** (2P/2Ref.^es^) for trust and sharing, helping to create a certain sense of **cheer/relief** (1P/1Ref.^e^) and **assurance of psychological well-being** (1P/1Ref.^e^), which **help**s **in difficult times** (1P/1Ref.^e^) and **help**ed in **facing** the end of life (1P/1Ref.^e^).

On the other hand, half of the FCs mentioned aspects that could potentially hinder the patient’s adaptation, such as the **conspiracy of silence** (4P/4Ref.^es^), which is associated with the difficulty FCs and patients experience when attempting to clearly and spontaneously address the questions linked to end-of-life care.

### 3.3. FCs’ Adaptation to Patients’ End of Life

Regarding the **FCs’ adaptation** to patients’ end of life (Table 3), the participants underlined the importance of the intervention in helping their own adaptation. The majority stated that the intervention **helped** in **facing** (5P/7Ref.^es^) the end-of-life stage of their loved one, contributing to their adaptation and to **prognosis acknowledgment** (2P/2Ref.^es^). Additionally, the intervention was equated to a **private space** (1P/1Ref.^e^). However, two participants revealed that the intervention **did not promote adaptation**, stating that it **did not help** (2P/2Ref.^es^); therefore, the **need for parallel support** (1P/1Ref.^e^) was mentioned.

### 3.4. The Role of the LL in the End-of-Life Adaptation Process

There was heterogeneity in the participants’ experiences regarding the reception of the LL. Two of the FCs received the letter during the **patient’s life** (2P/3Ref.^es^), and two FCs received the LL directly, meaning that it was **handed** over **by the patient** (2P/2Ref.^es^). One participant **found** (1P/2Ref.^es^) the LL unexpectedly after the patient’s death, and in another case, the patient indicated the place where the letter was to the FC so that it could be read *“(…) in case something happened”* (P1).

Seven FCs reported that they have read it, distinguishing three distinct reading moments: **before passing** (2P/2Ref.^es^), **right after passing** (4P/4Ref.^es^), and in the **post-mortem period** (1P/1Ref.^e^). In the **right after passing** moment, the FC had instructions from their loved ones to read the LL. Most of FCs mentioned that they read the LL **more than once** (5P/6Ref.^es^)—*“(...) this letter and until today it is kept on my bedside table, and I go there repeatedly to see it...”* (P1)—and some have read it only **once** (2P/3Ref.^es^).

Two participants explained that the **reason** (2P/2Ref.^es^) why they read the LL after the patient’s death relates to the fact that the LL lends a permanence to the voice of the loved one: *“(...) I sometimes read the letter. Looks like I’m listening to her saying that.”* (P5).

Nonetheless, there was evidence of **not** (1P/1Ref.^e^) reading the LL, despite **several reading attempts** (1P/1Ref.^e^). The key **reasons** for why the LL has not yet been read include the **fear of goodbye** (1P/2Ref.^es^)—*“(…) I didn’t want to know that because for me my father was there and if I read that I was afraid that he was saying goodbye to me while he was here and that was what mattered to me, he was with me and everything is fine.”* (P8)—and the **uncertainty in the time of death** (1P/1Ref.^e^).

Regarding the role of the LL in **patient**’s end-of-life adaptation (Table 4), the FCs noted that the LL allows them to express **what was being felt** (1P/2Ref.^es^) and that it was **revealing of future plans** (1P/2Ref.^es^).

Regarding the **importance** of the LL in the FCs’ process of adapting to the loss, it appears that the letter was, indeed, important for the majority of the FCs, with many answering **yes** (6P/7Ref.^es^) to a question about whether the LL was beneficial and one participant stating that it **helped** (1P/1Ref.^e^).

The **content** of the LL allows for the **reinforcement of what was transmitted in life** (1P/2Ref.^es^), as it **mirrors what** the sick family member **felt** (1P/1Ref.^e^), the **feeling towards the family** (1P/1Ref.^e^), and reflects the **mother’s character** (1P/1Ref.^e^). Although, one FC was **afraid of its content** (1P/1Ref.^e^).

In terms of **meaning**, the meaning attributed to the LL among the FCs seemed to vary; most of them highlighted its **importance for future generations** (6P/7Ref.^es^) and the LL’s **importance for the family** (6P/6Ref.^es^). Two of FCs mentioned its **importance for the FC** (2P/2Ref.^es^). In addition, they emphasize the **affective relationship** (2P/4Ref.^es^) that is described in the LL, the **satisfaction** with receiving the letter (2P/3Ref.^es^), and the **emotion** they felt when reading it (2P/2Ref.^es^). The letter itself “**allowed proximity**” (1P/1Ref.^e^) to the patient in life, enhancing the promotion of a **feeling of comfort** (2P/2Ref.^es^), facilitating **remembering and helping with longing** (1P/1Ref.^e^).

### 3.5. Grieving Process

The majority of participants reported that they did **not** receive **psychological support** (6P/6Ref.^es^) as they did not feel that they needed it during the grieving process (Table 5). It is worth noting that one participant revealed that he felt the **need for support** (1P/1Ref.^e^). On the other hand, two participants (P1, P7) received psychological support (**yes**—2P/2Ref.^es^) in the months following their loss, and one of them indicated that it **made the** grieving **process** of her mother **easier** (1P/1Ref.^e^).

When questioned about **post-loss difficulties**, all participants mentioned difficulties. They predominantly mentioned **absence** (2P/3Ref.^es^); **longing** (2P/2Ref.^es^); the **absence of guidance** (1P/1Ref.^e^); the **absence of company** (1P/1Ref.^e^); **difficulty in going to the graveyard** (1P/1Ref.^e^); and the difficulty in **dealing with other family members’ suffering** (1P/1Ref.^e^).

At the time of the interviews, when the FCs referred to their current emotional state, they presented a variety of emotions. The participants indicated that there was a **longing** feeling (3P/5Ref.^es^—P4, P5, P8), as well as a sense of **missing** (1P/1Ref.^e^) the loved one, and one FC admitted that they **think about her** (1P/1Ref.^e^). The FCs who lost their family members longer ago expressed more positive feelings, such as the sense of a **mission** being **accomplished** (3P/4Ref.^es^—P4, P5, P8); a **feeling** of tranquility/**calm** (1P/1Ref.^e^), and feeling **good** (2P/2Ref.^es^—P4, P6). Those who lost their relatives more recently expressed negative feelings, including the perception of not having overcome the situation (**did not get over**: 1P/1Ref.^e^), with one participant admitting that they are experiencing a certain **denial of reality** (1P/1Ref.^e^), despite being in the process of **acceptance** (1P/2Ref.^es^), resulting from the loss of her husband (5 months ago). Despite having lost his father 2 years and 5 months ago, another participant mentioned feeling as though they were under the influence of **emotional anesthesia** (1P/2Ref.^es^). Although this participant’s speech conveyed the notion of resolved parental grieving, the inconsistencies in their verbal and non-verbal speech pointed to a process still in progress.

There are several points that allow us to understand what **helps** FCs when **facing loss**, including the feeling of a **mission** being **accomplished** (2P/5Ref.^es^); the **family** (4P/4Ref.^es^) and the support it offers; **reason and finitude** (1P/2Ref.^es^); the guarantee of a **re-encounter** (1P/2Ref.^es^); the importance of **psychology appointments** (1P/1Ref.^e^); and **space reorganization** (1P/1Ref.^e^).

## 4. Discussion

This study aimed to understand the impact of MLT on the grieving process of the FCs of cancer patients who underwent the intervention and have already died.

The interviews showed that all the participants remembered having taken part in the previous investigation and their relatives having undergone MLT. The FCs believe that the intervention was, overall, a positive experience for the patient (P1, P2, P3, P4, P5, P6), promoting their adaptation and, in some situations, helping them to recognize their prognosis (P2, P4, P8). This is an important aspect, as awareness of the end-of-life prognosis is beneficial for patients [58] and can favor a more positive psychological experience of the disease [59]. In addition, they stated that it helped in the “most difficult moments” (P4) to deal with the end-of-life challenges, a result that is in line with other studies that have looked at the difficulties faced by patients during the end-of-life stage (e.g., Bruce et al., 2011 [6]; Onishi, 2021 [20]). The communication promoted by MLT was an element highlighted by the participants (P4, P5, P6), as MLT was seen by patients to be characterized by privacy, sharing, and almost exclusive trust [20,31,35]. It is understood, therefore, that the subjects addressed in the sessions promoted in the patient a feeling of cheer or relief, which the FCs recognized as a buffer in the adaptation process. In this context, the support provided will have contributed to the psychological well-being of the patients, at least at that time, helping to alleviate the psychosocial and existential suffering often experienced by terminal cancer patients [30,60]. Existential concerns are recognized as profoundly important for patients facing terminal illnesses [6,38], with a key aspect of palliative care centering around addressing the emotional, psychological, and communicational needs of patients with advanced and incurable cancer [61].

One aspect that deserves our attention is the conspiracy of silence, which was mentioned by half of the participants (P2, P5, P6, P7), despite patients having an insight into their disease prognosis (an inclusion criterion for MLT). In this context, we seem to be facing a double conspiracy, a pact of silence [62]. Counterproductive behavior on the part of FCs, either to protect the patient or for self-protection [8,9,62], was mentioned by the participants as having contributed to a lack of communication about important end-of-life aspects (e.g., saying goodbye, resolving pending issues, future directions). In addition, this type of behavior may have prevented patients from communicating openly about death and finitude and cooperating and collaborating with the phenomenon, an anxiogenic element for the patient [62,63]. In this context, MLT provided an opportunity for frank and open communication on the part of the patient, appearing as a communicative “window” for emotional ventilation (P5, P6), enhancing family sharing and communion.

With regard to the role played by MLT in the adaptation of FCs to the end of their loved one’s life, most of the participants considered that its contribution was positive (P1, P4, P6, P7, P8). The recollection of the actions associated with MLT (e.g., “going to appointments”, “answering questions”) and the identification of the conspiracy of silence in half of the participants leads us to question whether some of these answers were induced by the way our question regarding this was formulated, or whether these answers were products of “social desirability”, since there were difficulties in distinguishing between the process of adapting to the loved one’s end of life as a whole and, more specifically, the role of MLT in this process. However, aspects of MLT were pointed out as having been a facilitating element in the recognition of the prognosis of the patient’s health state and contact with health professionals (P8), allowing for the venting of emotions that are difficult to address with the patient and family members and the opportunity to look at a certain “problem” a family member may have during the end-of-life stage from another perspective. Accompanying a family member with cancer in the palliative phase is associated with suffering [1,20], being a difficult period that affects not only the patient but also the nucleus that surrounds them, sometimes requiring the intervention of professionals to lessen the shock and address the disease suffering [64]. This aspect was mentioned by one of the participants (P5), verbalizing the need for professional accompaniment parallel to the intervention being carried out with the patient.

We found that the LL has an important role in the family members’ grieving process and that the reception of the LL in life was avowedly considered as pleasant and positive. Most of the participants have read the LL several times (P1, P3, P4, P5, P6), validating the notion that the LL is a feature of a lasting legacy [65,66]. There were very positive references to the reading of the letter, which was seen to induce sensory experiences—*“I can hear her voice”* (P1)—and yield a closeness akin to “physical proximity”, constituting a comforting inner presence and an opportunity to say goodbye that was not possible in life (P5). In this context, the LL appears as a mental representation of the deceased family member, serving as a safe base and refuge [67], important elements in adapting to the loss of a loved one.

The FCs also mentioned the feeling of comfort provided by the LL in moments of longing and sadness (P1, P4) insofar as it reflects the character of the loved one, referring to the period before health decline (P1). This element is important, considering the multiple losses that occur in the process of dealing terminal illness, encompassing the loss of physical and psychological integrity, the loss of autonomy, and the erosion of one’s own identity [14], all of which can sometimes be felt more intensely by family members than by the patient themselves [68]. In this context, the LL offers the chance to recover and preserve the person’s identity, sometimes to the detriment of the “identity” of the patient. For one participant (P1), the comfort provided by the words written in the LL was so impactful that it prompted them to get a tattoo of one of the sentences in the letter as a symbol of the permanent mark of the deceased loved one [69]. This brings us to the lifelong nature of the document; as an autobiographical element [22] and a tangible object [31,36,45], the LL allows for the maintenance of emotional closeness, the preservation of identity, and the formation of a meaningful connection with the past [67].

In dyads where the existence of the conspiracy of silence was identified, the LL paradoxically appeared to be a means of “communication” between patients and FCs (P2, P6). In the absence of an open discourse on illness and death, its content allowed FCs to feel valued and grateful for the support and care provided to their family member at their end of life, contributing to the “mission accomplished” feeling, a protective factor in the grieving process. According to the literature, silent patterns cause communication blocks in the family system [62], and it seems that this was a way for the patient to establish, in life, communication with their family, highlighting the structuring and communicative nature of the LL as a central element of MLT [66] and also favoring adaptation in the grieving process.

The FCs (P1, P2, P4, P5, P6, P7) therefore recognized the importance of the LL in helping them to adapt to their loss. Its content is interpreted as reinforcing the information transmitted in life by patients about their character, their feelings towards the family, and the reflection of their emotions, validating the main themes identified in previous studies when analyzing the LLs of cancer patients at the end of their lives [44,66].

There were numerous references to the significance of the LL, which highlight its importance, constituting a living memory of the loved one written in the first person, emphasizing the transgenerational nature of the document [37,69,70,71]. The reinforcement of the affective relationship justifies the satisfaction reported by the FCs when reading the LL (P5, P6), given the need to perpetuate the bond with the deceased relative, as mentioned by Stroebe et al. (2010) [72].

The results of the present study reveal that all the FCs reported difficulties after their loss, with the absence of their loved one (P2, P5), the repercussions of their loss (lack of guidance, company, and difficulty in dealing with other family members’ suffering), and longing (P4, P8) all being mentioned, all of which can be expected while grieving, as they are characteristic of the grieving process [73,74,75,76].

Regularly and ritualistically visiting the cemetery of a loved one is a frequent behavior in the Portuguese population, and this was identified by one participant as a way of dealing with longing (P4). Contact with death impels the bereaved to adopt coping strategies [77], key elements of the adaptation to grief [78], such as (re)visiting their loved one’s “last address” (this is a figurative expression often used in the Portuguese language to refer to the tomb in the graveyard). This behavior can be interpreted as a loss-oriented strategy [75] which allows the bereaved to maintain a continuous connection the deceased family member [67] and preserve the significant bond they had with the deceased family member [72]. However, grief is not a static and linear process; rather, it is a dynamic and oscillating one wherein the bereaved person can resort to both confrontation and avoidance strategies [72,77].

The feeling of a mission being accomplished was the element most referenced by the FCs as having helped them face their losses, even in situations characterized by the conspiracy of silence. The feeling that everything possible had been done for the family member, shaped and confirmed by the LL, provided one of the participants (P6) with a feeling of well-being, including in relation to the decision to conspire, in line with the results obtained by Cejudo López et al. (2015) [9]. Family, as a social support network [2,69,79], is also perceived as a facilitator in this process (P2, P3, P4, P7), constituting a protective factor [46] with a buffer function [80,81], helping to mitigate the risks associated with the loss of a loved one [76,81,82]. The awareness of a loved one’s finitude (P5) and the expectation of a post-death reunion (P7) also appear to be strategies used by the participants to face the loss, allowing them to give it a new meaning [74,83].

When asked how they feel at the current moment, longing and the feeling of a mission being accomplished (P4, P5, P8) were the most mentioned and expected emotions, as referenced above. There were reports of a positive experience (e.g., acceptance and tranquility), in line with acceptance, the last stage of the grieving process [84], and with the fact that most of the participants had lost their relatives more than two years ago. Nevertheless, there were also testimonies that suggest that the FCs have difficulties facing their losses (e.g., emotional anesthesia and denial of reality). These difficulties, which do not always fit into a diagnosis of complicated grief [85,86], make integration and acceptance of the loss difficult despite adaptation. This was the case of one participant (P8) who admitted that they still do not feel ready to read the LL (which was given to her by her father six months before his death) and that they have difficulties dealing with the anticipation of the loss, a fact that has remained after his death. She presented justifying rationalizations that reveal these difficulties, consubstantiated in the strangeness and fear of becoming a “cold person”, despite the verbalization of a positive grief resolution, a concrete farewell, and the feeling of a mission being accomplished. This variability in experiences brings us back to the idiosyncrasies of the grieving process and the oscillating dynamics therein, which are sometimes reflected in confrontation and sometimes reflected in avoidance.

Most of the participants did not receive any psychological support after their loss (P2, P3, P4, P5, P6, P8), nor did they feel the need to. However, one of the participants (P5) consistently stressed the need for and relevance of professional accompaniment when they were accompanying the patient. This statement reminds us of the need for greater family member involvement in MLT, an area that Dignity Therapy has evolved in [1,13,87,88].

### 4.1. Implications for Clinical Practice

The results are clear regarding the existence of the conspiracy of silence, the implications of which are well known according to the scientific literature. In this way, it is important to make health professionals aware of this problem in the sense that they do not condone it when requested by family members, promoting, on the contrary, frank and open communication between patients and family members. Facilitating emotional expression and bringing agents closer together in this delicate process of advanced, progressive, and incurable disease serves to avoid the experience of silence and suffering in end-of-life processes. Approaching death as an integral part of life and human development is equally important, considering the difficulty at a social level in discussing end-of-life issues, which is obviously reflected in the way patients and their families deal with this issue when faced with the imminence of death.

MLT proved to be auspicious in promoting the adaptation of end-of-life patients, and the potential of MLT was also recognized by the FCs. MLT has potential because it offers a private space for cancer patients to express themselves emotionally and also because it seems to open up a window of communication between the dyad, breaking the “knots” of the conspiracy of silence. In this way, it constitutes a new structured psycho-existential intervention, a relevant resource to help health professionals to adequately respond to the holistic needs of patients in palliative care [44].

On the other hand, the LL resulting from this intervention process proved to be a central element in the process of adapting to the loss of the FCs. The life-long, transgenerational character of the LL promotes the maintenance of the bond and emotional closeness with the deceased loved one and mitigates longing, as emphasized by the FCs, reinforcing the relevance of the clinical use of this legacy document in the life review process, as well as the farewell process, and supporting families in the grieving process.

Finally, it is important to highlight the idiosyncrasies of the end-of-life processes and adaptation to loss, evident in our results, which refer to the importance of taking into account the singularities of patients and FCs in clinical interventions, particularly in MLT.

### 4.2. Limitations and Future Research

Our results show the relevance of integrating FCs into the interventions of end-of-life patients, helping healthcare professionals to respond in a sensitive and individualized way to the holistic needs of patients and their families in these circumstances. In this context, MLT must evolve to protect the patient–family dyad, taking into account the communication barriers identified earlier, which can result in frustrated farewells and lonely deaths. The positive results derived from integrating family members into Dignity Therapy, substantiate the applicability of family member integration to MLT [1,13,88].

Regarding the qualitative methodology we used in this study, it allows us to analyze a complex issue such as terminal illness in greater depth. However, scripts that guide the conduct of interviews require a very rigorous critical analysis in order to avoid questions that induce socially desirable responses. Conducting studies using a mixed methodology (qualitative and quantitative) may prove to be an asset in continuing to evaluate the effectiveness of MLT.

## 5. Conclusions

MLT was identified as a promotor of an environment conducive to patient privacy and emotional ventilation, signaling its potential in this area. The recognition of the role played by MLT in improving the psychological well-being of FCs’ relatives with cancer at the end-of-life stage and also in their acceptance of the departure of their family member, reflected in the adaptive course of the grieving process, is indicative of the effectiveness and relevance of this intervention in end-of-life care.

In our study, the resulting LL emerged as a tool that potentially promotes adaptive grieving as a communicational element, contributing to the grieving process insofar as it constitutes an autobiographical [22] and transgenerational document [37,69,70,71], a perpetual legacy that promotes a sense of comfort [39,70], strengthens relationships and closeness, and allows for the mitigation of longing.

## Figures and Tables

**Figure 1 healthcare-12-00471-f001:**
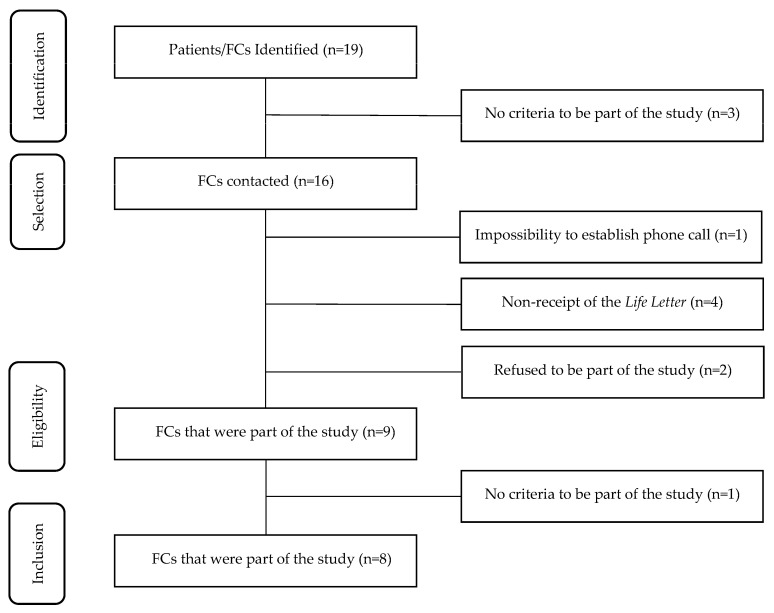
Sample selection flowchart.

**Table 1 healthcare-12-00471-t001:** Sample characterization.

		*n*	%	*M*	*SD*
Gender	Female	7	87.50		
	Male	1	12.50		
Age				38.38	11.86
Patient’s degree of kinship	Father/Mother	6	75.00		
	Husband/Wife	1	12.50		
	Granddaughter	1	12.50		
Time elapsed since death	Less than 1 year	1	12.50		
	1 to 2 years	1	12.50		
	2 to 3 years	3	37.50		
	More than 3 years	3	37.50		

**Table 2 healthcare-12-00471-t002:** Theme 2: “FCs’ Perceptions about Patients’ Adaptation to the End-of-Life Stage”.

Categories and Subcategories	P ^1^	Ref.^es^	Excerpt	
Promoted adaptation				
Positive	6	6	“*I… honestly I only see positive points (…)*” (P1)	
Prognosis acknowledgment	3	4	“*Although he knew, he had the notion that… of what was going on*” (P4)	
Private space	2	2	“*(…) he even told me that he liked it because he talked about things he didn’t talk about, that he hadn’t talked to anyone ….*” (P5)	
Partially	2	2	“*In a way, yes, although knowing his personality, I know that he was always going to be like this, that he was always going to react this way*” (P2)	
Believing so	1	1	“*I believe so… yes, I believe so.*” (P1)	
Help in difficult times	1	1	“*It always helps in these difficult moments, because a person is very sensitive and a little… desperate, isn’t it?*” (P4)	
Helped facing	1	1	“*Yes, it always helps a bit, isn’t it? … Because talking… it always helps a bit (…)*” (P4)	
Cheer/relief	1	1	“*(…) I remember that at least he left a bit more… cheerful (…), relieved, yes….*” (P8)	
Assurance of psychological well-being	1	1	“*I mean I want to believe that it psychologically helped him keep some sanity … (emotional) until the end.*” (P8)	
Did not promote adaptation				
Conspiracy of Silence	4	4	“*My father was never one to show himself to be “weak”, he always wanted to be strong for the family, he always hid this from my mother. He always… every letter, every exam he received he never showed my mother (…)*” (P2)	

^1^ Number of participants.

**Table 3 healthcare-12-00471-t003:** Theme 3: “FCs’ Adaptation to Patients’ End of Life”.

Categories and Subcategories	P ^1^	Ref.^es^	Excerpt
Promoted adaptation			
Helped facing	5	7	“*Yes, of course, it always helped.*” (P4)
Prognosis acknowledgment	2	2	“*Because actually I have always known what would happen to her.*” (P5)
Private space	1	1	“*I remember the times when I spoke with the Doctors at the time (…) I had another perspective of looking at the problem, which was (…) very big in my life. (…) and unburden too (…) and maybe think things differently. There were things I didn’t talk about, maybe with my mother or with other people.*” (P8)
Did not promote adaptation			
Did not help	2	2	“*To me it didn’t (a bit emotional).*” (P2)
Need for parallel support	1	1	“*I believe at the time me as a daughter… maybe having a session… being with the Psychologist that followed her or… would have helped me.*” (P5)

^1^ Number of participants.

**Table 4 healthcare-12-00471-t004:** Theme 4: “The Role of the LL in the End-of-Life Adaptation Process”.

Categories e Subcategories	P ^1^	Ref.^es^	Excerpt
For patients			
Revealing of what was being felt	1	2	*“It was important because (…) in that letter he expressed a little more about what he was feeling. Until then he never... I tried talking to him and he never, never expressed much.”* (P2)
Revealing of future plans	1	2	*“And there in the letter, right, the way he talks… about a lot of things and everything, maybe things that he wasn’t going to tell anyone because, in his future plans, he was meant to live, not to die (…)”* (P7)
For FCs			
Importance in adaptation			
Yes	6	7	*“Yes… it helped, it did. (…) That letter helped me a little to accept it, sure.”* (P2)
No	1	1	*“In that regard I won’t say (…) that the letter was important…. (…). It was important, but not in that aspect (…)”* (P3)
Helped	1	1	*“(…) I believe that… that at the time I didn’t quite understand the meaning of that, but after some time I realized what it meant, that she externalized what she felt, right? In several ways, and that has helped us all.”* (P1)
Content			
Reinforcement of what was transmitted in life	1	2	*“What he wrote is what he told me every day.”* (P7)
Mother’s character	1	1	*“Because that was exactly what my mother was, an honest person, of honour…”* (P1)
Afraid of its content	1	1	*“I wanted to, but I believe I was afraid. (…) of what was in there.”* (P5)
Feeling towards the family	1	1	*“(…) he explained there how much he cared for us, his family. He cared about family very much.”* (P4)
Mirrors what was being felt	1	1	*“What he wrote there is what he felt.”* (P7)
Meaning			
Importance for future generations	6	7	*“It can because he has small grandchildren that are still unable to understand life, right? And they… (…) they know his grandfather but, of course, as the years go by that… that can be lost and I believe it’s a beautiful memory of him (…)”* (P2)
Importance for the family	6	6	*“(…) my sisters even said “look we have to make copies so that each one gets, each one with the … (…) letter, with what was written in it.””* (P3)
Affective relationship	2	4	*“And that she really liked me. I knew it, but… the… seeing it in the letter, having been told to someone else who wrote it...”* (P5)
Satisfaction	2	3	*“(…) and I was very pleased with it, absolutely…. I was very pleased with the letter.”* (P5)
Farewell	1	2	*“(…) I see in that letter almost a way for her to say goodbye to me, to…. Because we didn’t actually have that (…).”* (P5)
Emotion	2	2	*“(…) what I felt. Look, a lot of emotion.”* (P3)
Importance for the FC	2	2	*“It was, to me.”* (P2)
Feeling of comfort	2	2	*“(...) when, for example, when it’s been her death or her birthday... or when I feel a bit sadder it comforts a little…”* (P1)
Tattoo	1	2	*“I even tattooed her, I tattooed a sentence that is there.”* (P1)
Good relationship	1	1	*“Well… I felt that I actually had… I had a good relationship with her.”* (P5)
“It allowed proximity”	1	1	*“Actually, for me it was good, it was good because my father wasn’t someone that shared a lot of… his emotions and in that letter he showed a bit more and it was good, (…) it allowed to be a bit closer to him, let’s say.”* (P2)
Comfort and memory	1	1	*“(...) exactly what my mother was (...) and that comforts me. There are times in which due to the whole process we went through you know, of seeing her degrade day after day, we forget a little bit who she was, before having anything. And that reminds us.”* (P1)
Remembering and helping with longing	1	1	*“Well, I… I hold on to everything, right? That letter is good for when… (…) I miss her a lot (…)”* (P7)
Only memory	1	1	*“(…) it’s the only thing I keep from my mother, is that letter.”* (P1)

^1^ Number of participants.

**Table 5 healthcare-12-00471-t005:** Theme 5: “Grieving Process”.

Categories and Subcategories	P ^1^	Ref.^es^	Excerpt
Psychological Support			
No	6	6	*“No, no.”* (P4)
Need for support	1	1	*“(…) maybe I needed it at the time and I have a good family support that helped me a lot, but (…)… a talk would be very important*.*”* (P5)
Yes	2	2	*“I had an appointment a month ago (…)”* (P7)
Made the process easier	1	1	*“Let’s say that (…) it helped me not only in the process of my mother, but also in other problems (...). And me being well resolved with everything else facilitated the grieving process.”* (P1)
Post-loss difficulties			
Absence	2	3	*“(…) her not being there. And the things that are happening to us, she… not sharing with her. (…) the longing, me wanting to tell her, share something of mine with her, good, bad… and she is not there*.*”* (P5)
Only family member	2	3	*“It’s more... I have an uncle who is single and lived with her and, since my grandmother’s passing, he isn’t doing well at all and it is difficult, (...) he continues to live in the house that was my grandmother’s and it is very difficult to be there with him.”* (P6)
Longing	2	2	*“I feel… I miss him every day, I wish I had him here (…)”* (P8)
Absence of guidance	1	1	*“I think it’s not having her approval (...) sometimes I feel lost (...)”* (P1)
Absence of company	1	1	*“The company, the advices when I get home, always having that friendly word… Going home to be with him.”* (P7)
Difficulty in going to the graveyard	1	1	*“I don’t feel like going to the cemetery, but I never liked cemeteries. And it’s a little difficult for me to go (...). And I think it’s a bit difficult for me to divide his emotional presence that I have at home and... physical that I think he doesn’t even want to be there anymore. But knowing that is where he, where he is.”* (P8)
Dealing with other family members’ suffering	1	1	*“(…) my mother still doesn’t accept it. And it is very difficult, even living with my mother is very difficult, at this moment.”* (P2)
How one is feeling			
Longing	3	5	*“Look, right now… I miss her. Missing….”* (P5)
Mission accomplished	3	4	*“And it really helped me knowing that I always did the best with her. Everything that was, that was possible, that… that… there was nothing left to do with her.”* (P5)
Acceptance	1	2	*“I think I’m accepting it, I think I’m moving towards acceptance.”* (P7)
Emotional anesthesia	1	2	*“(...) I believe I’ve handled it well and I think it’s something I’m proud of myself, was to deal with... with all this that wasn’t easy. But I think I had... those 2 years of treatments and everything, I think, I think... they taught me something. (...) I think I... I don’t know, people say they get tougher or... No, but sometimes I’m afraid that I might start to become a cold person (...)”* (P8)
Good	2	2	*“I feel good, I accepted it well*.*”* (P6)
Missing	1	1	*“Because at this moment I miss her a lot (...)”* (P1)
Did not get over	1	1	*“I did not overcome it very well…. (…) It is difficult for me to talk about him.”* (P2)
Denial of reality	1	1	*“(…) sometimes I’m waiting for him to be discharged from the hospital.”* (P7)
“I think about her”	1	1	*“(…) I won’t say that I don’t think about her, I do, of course.”* (P3)
Feeling calm	1	1	*“(…) I felt calm for the first time.”* (P8)
Help facing the loss			
Mission accomplished	2	5	*“I think I was... I think I did everything I could have done for him. I was always present, I came to all the treatments, I went to all the appointments. We always talk about this. My father told me, I believe, everything he had to say to me. I told him everything I had to say too. (...) I strongly believe everything has been said, (...) I feel at peace, that’s it.”* (P8)
Family	4	4	*“What helps me is my family. I have two children, it’s being with my children. I rely a lot on them, on my wife (...)”* (P2)
Reason and finitude	1	2	*“People like me have an end and like… we all have an end don’t we. The reason turned out to be… we will pass the time and it will, things have to happen.”* (P5)
Re-encounter	1	2	*“Knowing that one day I will find him.”* (P7)
Psychology Appointments	1	1	*“Very honestly, appointments with (…) helped me a lot… a lot, without a doubt.”* (P1)
Space reorganization	1	1	*“When I’m at home, if I’m tidying up the house, I remember him (…). And I already moved all the furniture in the house. (…) It’s not so that I don’t feel… so that his chair isn’t empty.”* (P7)

^1^ Number of participants.

## Data Availability

The data presented in this study are not publicly available due to privacy restrictions.

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
