# Peer review of "Life beyond Loss: A Retrospective Analysis of the Impact of Meaning of Life Therapy on the Grieving Process of Cancer Patients’ Family Caregivers"

_healthcare, 2024, doi:10.3390/healthcare12040471_

Round 1

Reviewer 1 Report

Comments and Suggestions for Authors

Line 41: Can another word be substituted for ‘agents’? It is unclear what you are referring to in this sentence, which are factors for dealing with palliative care of individuals.

Line 46: You refer to the conspiracy of silence. Defining this conspiracy or adding more information would be helpful. I’m not sure everyone reading will know what this is. Thank you.

Or include the next paragraph (line 48) as part of the previous. This would clarify the silence issue.

Line 449–450: Please add this one sentence to a paragraph.

Lines 486-492: repetitive. Please revise.

Lines 589–593: The last paragraph of the conclusion has new material (cited articles from references). Please review and revise to include only the material mentioned in the body of the manuscript.

The present study appears as an opportunity to ‘give voice’ to family caregivers of deceased cancer patients.

There is a large investment in monitoring patients and their relatives in the process of palliative and terminal cancer disease; however, the authors feel there is a gap in the post-mortem follow-up of family caregivers.

The following objectives were defined:

1) To know the family caregivers (FC) perception of the impact of MLT on the patients’ adaptation to the End-of-Life;

2) To know the FC´s perception of the impact of MLT on their relative's adaptation to their

impending End-of-Life  

3) Evaluate the role of the living letters (LL) in the grieving process; 

4) Explore the adaptation in the grieving process.

The added value of this project reflects the family caregiver’s perception of the value of MLT while their relative was alive. During this intervention, the cancer patient wrote a Living Letter as a means of relieving some of the grief after their death. Limited literature on this specific approach is available to understand grief and how individuals process the event.

The conclusions are mostly consistent with the body of the manuscript; however, what could be considered new material in the last paragraph needs to be reviewed and revised.

References are appropriate; however, there are thirteen literature reviews (15%). Most of the dates are within the five-year range. Those that are older refer to the historical information necessary for background and are helpful for the reader's understanding of the therapy the authors developed.

Comments on the Quality of English Language

Lines 39–44, 53–75, and 413–430: These paragraphs have long sentences that need to be shortened into two or more for clarity. Please review and revise, and review and revise in other places in your manuscript.

Lines 143–150: Please review and revise. Check spelling and grammar (verb tense). This is difficult to follow.

Author Response

Dear Reviewer,

Reviewer 2 Report

Comments and Suggestions for Authors

Thank you very much for offering this opportunity to review this manuscript. I enjoyed reading it. The purpose of this study is “to understand the impact of Meaning of Life Therapy on eight family caregivers grieving process and their perception about the role of the life letter on grief. The manuscript is generally well written; reflects rigorous methodology; and contributes to the understanding the role played by Meaning of Life Therapy in improving the psychological well-being of both cancer patients and their family caregivers.

My only minor comment is that it may be proper to present the full names of the abbreviation, when it is applied in the first time, e.g., FCs and family caregivers (the full names were applied under introduction section).

Author Response

Dear Reviewer,

Reviewer 3 Report

Comments and Suggestions for Authors

Dear Authors

I have reviewed the manuscript entitled “Life beyond Loss: A Retrospective Analysis of the Impact of Meaning of Life Therapy on the Grieving Process of Family Caregivers of Cancer Patients”. It is a longitudinal study developed in north of Portugal. The sample of this study were 8 FCs of cancer patients (n=19) submitted to Meaning of Life Therapy (MLT) and elaborated the ‘Life Letter’ (LL) due MLT. The aim of this study was: to understand the impact of MLT on eight FCs grieving process and their perception about the role of the LL on grief, using semi-structured interviews. Through qualitative analysis suggested that MLT was perceived by FCs as a positive experience, despite the conspiracy of silence identified. The LL was interpreted as a communicational element, promoting emotional closeness with the cancer patients, being a valuable tool in the FC’s adaptation process of loss.

The study is very interesting and innovative. The MLT is a dignity way to allow dying people to leave legacy for FC´s. The outcomes are especially important in palliative care, namely in cancer end-of-life patients who had a very high level of suffering. The thematic falls within with special issue “Psychological, Cognitive and Physical Dimensions of Adjustment to Cancer: Perspectives of Cancer Survivors and Caregivers”.

I have some suggests to improve your paper:

11)     Line 141 (page 3): it is described the inclusion criteria in point 2 “The LL has already been received by FC” and in diagram page 4: is mentioned in second exclusion “Non-receipt of the Life Letter” isn´t should be already excluded in the first exclusion? as it does not meet inclusion criteria?

22)     In table 1: change “N” by “n”;

33)     Is not clear if the interviews were developed by health professionals, because concerning the questions content, could be sensible to predict some psychological interventions if identified during the interview (as ethical concerning).

44)     Did you make some previous validation of your data instrument?

55)     It could be interesting to have a paragraph explain who you recruited patients? (from a specific clinical appointment?) the inclusion end exclusion of cancer patients in MLT and some descriptions about the MLT phases to help readers that could not be familiar with the therapy.

It was delighting read your paper.

I have nothing to add and I wish you good luck towards publishing it!

Best regards.

Author Response

Dear Reviewer,

Reviewer 4 Report

Comments and Suggestions for Authors

Thank you for the opportunity to review this manuscript. Interesting and informative, well written and presented.

My only comments are:

from line 251 onwards remove bold

express within text of manuscript ethical approval was approved. 

Author Response

Dear Reviewer,
